# Enhanced Photo-Fenton Activity of SnO_2_/α-Fe_2_O_3_ Composites Prepared by a Two-Step Solvothermal Method

**DOI:** 10.3390/ma15051743

**Published:** 2022-02-25

**Authors:** Pinghua Li, Xuye Zhuang, Jiahuan Xu, Liuxia Ruan, Yangfan Jiang, Jiaxin Lin, Xianmin Zhang

**Affiliations:** 1School of Mechanical Engineering, Shandong University of Technology, Zibo 255049, China; lipinghua@sdut.edu.cn; 2State Key Laboratory of Applied Optics, Changchun Institute of Optics, Fine Mechanics and Physics, Chinese Academy of Sciences, Changchun 130033, China; 3School of Science, Jiangsu University of Science and Technology, Zhenjiang 212100, China; jiahuanxu@just.edu.cn; 4School of Material Science and Engineering, Northeastern University, Shenyang 110819, China; 1710168@stu.neu.edu.cn (L.R.); 1870323@stu.neu.edu.cn (Y.J.); 1800508@stu.neu.edu.cn (J.L.)

**Keywords:** hydrothermal method, heterogeneous composite, morphology, photo-Fenton activity, photodegradation

## Abstract

The x-SnO_2_/α-Fe_2_O_3_ (x = 0.04, 0.07, and 0.1) heterogeneous composites were successfully prepared via a two-step solvothermal method. These composites were systematically characterized by the X-ray diffraction technique, field emission scanning electron microscopy, an energy dispersive spectrometer, X-ray photoelectron spectroscopy and a UV–visible spectrometer. It was found that SnO_2_ nanoparticles were uniformly decorated on the surface of α-Fe_2_O_3_ particles in these heterogeneous composites. A comparative study of methylene blue (MB) photodegradation by α-Fe_2_O_3_ and x-SnO_2_/α-Fe_2_O_3_ composites was carried out. All x-SnO_2_/α-Fe_2_O_3_ composites showed higher MB photodegradation efficiency than α-Fe_2_O_3_. When x = 0.07, the MB photodegradation efficiency can reach 97% in 60 min. Meanwhile, the first-order kinetic studies demonstrated that the optimal rate constant of 0.07-SnO_2_/α-Fe_2_O_3_ composite was 0.0537 min^−1^, while that of pure α-Fe_2_O_3_ was only 0.0191 min^−1^. The catalytic mechanism of MB photodegradation by SnO_2_/α-Fe_2_O_3_ was examined. The SnO_2_ can act as a sink and help the effective transfer of photo-generated electrons for decomposing hydrogen peroxide (H_2_O_2_) into active radicals. This work can provide a new insight into the catalytic mechanism of the photo-Fenton process.

## 1. Introduction

As an important advanced oxidation process, the heterogeneous photo-Fenton system has been considered a promising method for the removal of stubborn organic dyes [1,2,3,4]. In this process, iron-based catalysts are generally applied to activate H_2_O_2_ in order to generate strong oxidative hydroxyl radicals (·OH) [5,6,7,8]. Among the iron-based catalysts, α-Fe_2_O_3_ is one of the most promising Fenton candidates due to its stable structure, low cost, wide absorption of visible light, and environmental benignity [9,10]. However, several adverse factors seriously reduce the reaction activity of α-Fe_2_O_3_, such as the high recombination rate of photoelectrons and holes, and a weak activation in alkaline environments. To remedy these drawbacks, various measures have been studied, such as porous regulation [11,12,13,14,15,16], facet engineering [17,18,19,20], and composite construction [21,22,23,24,25,26]. Among these methods, integrating α-Fe_2_O_3_ with other catalysts has caught the attention of many because it can effectively separate photo-generated electron–hole pairs. Liu et al. [27] reported the synthesis of α-Fe_2_O_3_ anchored to a graphene oxide nanosheet. The graphene oxide was considered to accelerate the transfer of photo-generated electrons and to enhance the absorption to methylene blue (MB) through electro-static interaction and π–π stacking. Deng et al. [28] constructed an advanced TiO_2_/Fe_2_TiO_5_/Fe_2_O_3_ heterojunction structure, and the abundant phase interfaces improved both the migration and separation of charges.

SnO_2_ has a high photochemical property and stability, and is extensively studied in the fields of Li-ion batteries [29,30,31], dye sensitized solar cells [32,33,34], and gas sensors [35,36,37]. Additionally, many studies have shown that the SnO_2_/α-Fe_2_O_3_ heterogeneous catalyst has an excellent photodegradation activity. Wang et al. [38] synthesized SnO_2_-encapsulated α-Fe_2_O_3_ nanocubes by annealing Prussian blue microcubes, and revealed the important contribution of SnO_2_ cubic shells for improving photocatalytic performance. Tian et al. [39] prepared a tube-like SnO_2_/α-Fe_2_O_3_ heterostructure by using an anion-assisted hydrothermal route and studied the effective separation of photo-generated carriers. Niu et al. [40] synthesized branched SnO_2_/α-Fe_2_O_3_ composites by a hydrothermal system of Sn(OH)_6_^2−^ dilute aqueous solution, and investigated their photocatalytic activity. The synthesis methods can significantly influence the morphology of SnO_2_/α-Fe_2_O_3_ composites. First, the SnO_2_ and α-Fe_2_O_3_ were prepared by a sol–gel method, then their composite (SnO_2_–α-Fe_2_O_3_) systems were synthesized by combining SnO_2_ with α-Fe_2_O_3_ in various weight percent ratios, and finally, the photocatalytic activity was investigated [41]. A necklace-like SnO_2_/α-Fe_2_O_3_ hierarchical heterostructure was fabricated by the chemical vapor deposition method, using SnO_2_ nanowires with the preferential growth direction of [1] as a template, and then the photocatalytic property was studied [42]. Therefore, it is interesting to explore the new synthesis methods as well as establish the relationship between morphology and internal catalytic mechanism of SnO_2_/α-Fe_2_O_3_ composites. 

In this study, we synthesized the x-SnO_2_/α-Fe_2_O_3_ (x = 0.04, 0.07, and 0.1) heterogeneous catalysts via a two-step hydrothermal method. The materials were systematically characterized by using the X-ray diffraction (XRD) technique, field emission scanning electron microscopy (FE-SEM), X-ray photoelectron spectroscopy (XPS), and a UV–visible (UV–vis) spectrometer. The XRD analysis indicated the successful synthesis of purity and well-crystalline of SnO_2_/α-Fe_2_O_3_ powders. FE-SEM images showed that SnO_2_ nanoparticles were homogeneously decorated on the surface of the peach-like α-Fe_2_O_3_. XPS measurement demonstrated Fe and Sn elements in the 0.07-SnO_2_/α-Fe_2_O_3_ sample to be trivalent and tetravalent, respectively. The UV–vis spectra revealed the strong visible light absorption ability of SnO_2_/α-Fe_2_O_3_ powders. The photodegradation of MB over different catalysts was tested, and the first-order kinetic analysis was performed to get the photodegradation rate. Free radical trapping experiments and hydroxyl radical quantitative experiments were carried out to explore the mechanism of photocatalytic reaction.

## 2. Experimental Section

### 2.1. Materials and Chemicals

Ferric chloride hexahydrate (FeCl_3_·6H_2_O) (Macklin, Shanghai, China), tin chloride pentahydrate (SnCl_4_·5H_2_O) (Macklin, Shanghai, China), urea (CO(NH_2_)_2_) (Macklin, Shanghai, China), polyethylene glycol (PEG) (Sinopharm, Shanghai, China), sodium hydroxide (NaOH) (Sinopharm, Shanghai, China), ethanol (C_2_H_5_OH) (Sinopharm, Shanghai, China), and methylene blue (MB) (Sinopharm, Shanghai, China) were used in the experiment. All the reagents were analytical grade and used as received without further purification. Deionized water was used throughout the experiment. 

### 2.2. Synthesis

#### 2.2.1. Synthesis of α-Fe_2_O_3_

The peach-like α-Fe_2_O_3_ powders were prepared by a facile hydrothermal method, as shown in the upper section of Figure 1. First, 1.623 g of FeCl_3_·6H_2_O, 0.6 g of polyethylene glycol, and 0.6 g of NaOH were dissolved in 60 mL of deionized water. Secondly, the mixture solution was stirred for 30 min and transferred into a 100 mL Teflon-lined stainless steel autoclave (H-100ml, Zhuoran Company, Zhengzhou, China). The thermal treatment was performed at 180 °C for 5 h. After the autoclave was naturally cooled down to room temperature, the precipitate was continually washed with deionized water and absolute ethanol. Finally, the α-Fe_2_O_3_ was achieved after being dried out at 60 °C for 2 h.

#### 2.2.2. Synthesis of SnO_2_/α-Fe_2_O_3_

The SnO_2_/α-Fe_2_O_3_ composites were prepared via a two-step hydrothermal method. In this method, the first step is used to prepare α-Fe_2_O_3_ precursors as in the above description. The second step is to grow SnO_2_ on the surface of α-Fe_2_O_3_ precursors as follows (shown in the bottom of Figure 1). The x g (x = 0.04, 0.07, and 0.1, which denoted the mass of tin chloride pentahydrate) SnCl_4_·5H_2_O, 0.5 g of CO(NH_2_)_2_ were dispersed in a mixed solvent consisting of 30 mL of deionized water and 20 mL of absolute ethanol. Then, 0.1 g of the α-Fe_2_O_3_ precursor was added into the above solution. Subsequently, the mixture solution was stirred for 10 min before transferred into a 100 mL Teflon-lined stainless steel autoclave. The autoclave was sealed and heated at 170 °C for 10 h and then naturally cooled down to room temperature. The obtained SnO_2_/α-Fe_2_O_3_ photocatalysts were washed several times with deionized water and absolute ethanol before they were dried at 60 °C for 12 h. The final products were termed as x-SnO_2_/α-Fe_2_O_3_ (x = 0.04, 0.07, and 0.1). The precise control of reaction temperatures is very important to obtain target samples with good crystal structures and homogeneous particle size [43]. Pure SnO_2_ can also be obtained by the process in the bottom of Figure 1 without adding the α-Fe_2_O_3_.

### 2.3. Characterization

Phase structures of powders were analyzed by XRD using Japan Rigaku (Tokyo, Japan) with a Cu target. The morphologies and chemical compositions of samples were characterized by FE-SEM (JEM-7001F, JEOL, Tokyo, Japan), equipped with an energy dispersive spectroscope (EDS). X-ray photoelectron spectroscopy of SnO_2_/α-Fe_2_O_3_ was detected using Thermo Fisher 250-XI (Thermofisher, Waltham, MA, USA) with an Al Kα. The optical properties and photocatalytic activities were measured using a UV–vis spectrophotometer (Shanghai Metash, UV-9000S, Shanghai, China). The molar ratios of composites were checked by an inductive coupled plasma emission spectrometer (Agilent ICP-MS 7500a, Santa Clara, CA, USA).

### 2.4. Photodegradation Experiment

The photo-Fenton activity of the samples was investigated by the degradation of MB dyes under visible light irradiation. The light source used was a 300 W xenon lamp with a 420 nm cutoff filter (Microsolar 300, PerfectLight, Beijing, China). In a typical procedure, 20 mg of as-prepared samples was dispersed into 100 mL of MB solution (20 mg·L^−1^) with the assistance of ultrasonic for 2 min. Afterwards, the reaction mixture was magnetically stirred in the dark for 30 min to ensure absorption equilibrium. The reaction was initiated by adding 1 mL hydrogen peroxide solution (H_2_O_2_, 30 wt%, Sinopharm, Shanghai, China) after the xenon light was stable. During the irradiation, 4 mL solution was sampled at 10 min intervals. The whole process took place under stirring, while the circulating cooling water worked at the same time. After removing the catalysts from each sample by centrifugation, the degree of photodegradation was calculated by measuring the absorbance of MB at 664 nm where the solution had the maximum absorption.

## 3. Results and Discussion

The crystal structure of as-prepared α-Fe_2_O_3_ precursors, SnO_2_, and x-SnO_2_/α-Fe_2_O_3_ (x = 0.04, 0.07, and 0.1) powders were measured by XRD. As shown in Figure 1, all the peaks appearing in the α-Fe_2_O_3_ powders are sharp and well indexed to a pure rhombohedral structure of hematite (JCPDS No. 33-0664). This indicates the high purity and good crystallinity of the prepared α-Fe_2_O_3_ sample. The widths of peaks of SnO_2_ are much larger than that of α-Fe_2_O_3_, which might be attributed to the poor crystallinity and small particle size of SnO_2_. After being decorated by SnO_2_, all the extra diffraction peaks of x-SnO_2_/α-Fe_2_O_3_ were indexed to (110), (101), and (211) planes of rutile phase of SnO_2_ (JCPDS No. 41-1445). This illustrates the successful synthesis of x-SnO_2_/α-Fe_2_O_3_ composites. The particle size was estimated using the Scherrer equation [44]: $D=\frac{K\lambda }{\beta \;COS\;\theta }$ where D is the grain size, K is the Scherrer’s constant, λ is the X-ray wavelength (0.154 nm), β is the FHWM, and θ is the diffraction angle. The grain sizes of SnO_2_, α-Fe_2_O_3_, and 0.07-SnO_2_/α-Fe_2_O_3_ are about 15 nm, 203 nm, and 212 nm, respectively.

The morphologies of α-Fe_2_O_3_ and 0.07-SnO_2_/α-Fe_2_O_3_ were measured by FE-SEM. As shown in Figure 2a, α-Fe_2_O_3_ is peach-like, and the inwardly concave symmetry curve can be clearly observed on the surface of α-Fe_2_O_3_. The α-Fe_2_O_3_ particle is homogeneous, with a diameter of around 220 nm. Interestingly, many small spots are decorated on the outer surface of α-Fe_2_O_3_ after adding Sn, as shown in Figure 2b. Moreover, those spots, each of around 20 nm, are uniformly distributed. The particle sizes essentially agree with the results from the above XRD, which were calculated using the Scherrer’s equation. Figure 2c indicates the EDS spectra of 0.07-SnO_2_/α-Fe_2_O_3_. It clearly identifies that the composite is composed of Fe, O, and Sn elements. Considering the XRD of 0.07-SnO_2_/α-Fe_2_O_3_, it is reasonable to assume that these small spots are SnO_2_. The chemical compositions of x-SnO_2_/α-Fe_2_O_3_ (x = 0.04, 0.07, and 0.1) were checked by ICP-MS. The mass contents of Fe_2_O_3_ and SnO_2_ in 0.04-SnO_2_/α-Fe_2_O_3_, 0.07-SnO_2_/α-Fe_2_O_3_, and 0.1-SnO_2_/α-Fe_2_O_3_ are 87.3% and 12.7%, 78.5% and 21.5%, and 70.6% and 29.4%, respectively.

To further demonstrate the compositions of surface dots and the oxidation states of metal elements of 0.07-SnO_2_/α-Fe_2_O_3_ powders, an XPS measurement was taken. Figure 3a shows the full-scale XPS spectrum, and the presence of Sn, Fe, O, and C was confirmed without any other element. The characteristic peak of C 1s found at 284.8 eV was from adventitious carbon. Figure 3b is the high-resolution XPS spectrum of Fe 2p. The binding energy peaks at both 710.5 and 724.5 eV were ascribed to Fe 2p_3/2_ and Fe 2p_1/2_, respectively. The peak for Fe 2p_3/2_ at 710.5 eV is sharper than that for Fe 2p_1/2_ due to the spin–orbit coupling [45]. The appearance of satellite peaks at 712.8 and 733.5 eV confirms that the Fe element in 0.07-SnO_2_/α-Fe_2_O_3_ is trivalent. A high intensity peak at the binding energy of 716.6 eV is due to the presence of Sn 3p_3/2_. Figure 3c shows the high-resolution XPS spectrum of Sn 3d. The peaks at 487.2 and 495.5 eV were attributed to Sn 3d_5/2_ and Sn 3d_3/2_, respectively. This could support the +4 oxidation states of SnO_2_. Thus, the results clearly confirm that these heterogeneous catalysts are composed of 0.07-SnO_2_/α-Fe_2_O_3_, agreeing with the XRD results in Figure 1.

In order to study the visible light absorption properties of α-Fe_2_O_3_, SnO_2_, and 0.07-SnO_2_/α-Fe_2_O_3_ samples, their UV–vis absorption spectra were measured, as shown in Figure 4. The present SnO_2_ could hardly absorb visible light (λ > 420 nm) in the UV–vis diffuse reflection spectrum, agreeing with the literature [46,47]. Interestingly, the 0.07-SnO_2_/α-Fe_2_O_3_ composite exhibited much higher photo absorption ability compared with pure α-Fe_2_O_3_. This might contribute to the improvements in the photocatalytic activities of 0.07-SnO_2_/α-Fe_2_O_3_.

The photo-Fenton activities of the present samples were studied using a degrading MB experiment under visible light for 60 min. One milliliter of H_2_O_2_ was added to the MB solution to activate the Fenton reaction. Figure 5a shows the visible light Fenton degradation of MB under different catalysts, including α-Fe_2_O_3_, SnO_2_, and x-SnO_2_/α-Fe_2_O_3_ (x = 0.04, 0.07, and 0.1) heterogeneous catalysts. Self-degradation of MB is limited. In the dark stage, SnO_2_ shows excellent adsorption of dye molecules, while α-Fe_2_O_3_ presents poor adsorption. The adsorption capacity of the x-SnO_2_/α-Fe_2_O_3_ composite was improved due to the decoration of SnO_2_. More adsorption means a closer contact between the catalyst and the dye molecules, which might contribute to the photo-Fenton reaction. Under visible light irradiation, 66% of the MB was degraded within 60 min in the presence of α-Fe_2_O_3_. Interestingly, the MB degradation of x-SnO_2_/α-Fe_2_O_3_ was much faster than that of α-Fe_2_O_3_. Moreover, with the increase in SnO_2_ dosage, the degradation efficiency increased and then decreased, reaching the optimal efficiency of 97% for the 0.07-SnO_2_/α-Fe_2_O_3_ sample. 

With a low concentration of MB, the degradation followed the pseudo-first-order kinetics and the reaction constant of photodegradation was determined by the following equation [28]:
$$\ln(C_{0}/\mathrm{Ct})=\mathrm{kt}$$
where C_0_ is the initial dye concentration which reached adsorption–desorption equilibrium in the dark, C_t_ is the dye concentration at given time t during the Fenton process, and k is the reaction rate constant. As shown in Figure 5b, the plots ln(C_0_/Ct) versus irradiation time are almost linear, which indicates that the photocatalytic degradation of MB solution agrees with the pseudo-first-order kinetic model. The optimal reaction rate constant was obtained for the 0.07-SnO_2_/α-Fe_2_O_3_ sample (0.0537 min^–^^1^), which was more than 2.8 times higher than that of α-Fe_2_O_3_ (0.0191 min^–1^). Thus, it can be concluded that SnO_2_ shows positive effects on the MB Fenton degradation of α-Fe_2_O_3_ [46,47,48]. Previous studies have reported that H_2_O_2_ could capture the photo-generated electrons of a semiconductor and decompose itself into ·OH for the dye’s degradation [49,50]. In this study, all the degradation efficiencies are enhanced with the assistance of H_2_O_2_ and the degradation efficiency of α-Fe_2_O_3_ is much higher than that of SnO_2_. These might be ascribed to the photo-Fenton reaction of Fe^2+^ and H_2_O_2_.

To assess the contribution of reactive radicals and further explore catalytic mechanisms, control experiments were carried out with or without scavengers over the 0.07-SnO_2_/α-Fe_2_O_3_ catalyst shown in Figure 6. Isopropanol (IPA) and 1, 4-benzoquinone (BQ) were used as hydroxyl radicals (·OH) and superoxide radical (·O_2_) scavengers, respectively. As shown in Figure 6, the catalytic degradation of MB was limited by scavengers. This indicates that both the ·O_2_ and ·OH have significant effects on the MB degradation of SnO_2_/α-Fe_2_O_3_. 

In addition, experiments on the amount of ·OH generation under α-Fe_2_O_3_ and 0.07-SnO_2_/α-Fe_2_O_3_ were performed to further explore the effect of SnO_2_ on the photo-Fenton reaction. The amount of ·OH in the heterogeneous photo-Fenton reaction was measured based on the fluorescence intensity of hydroxy terephthalic acid. As shown in Figure 7, the concentration of ·OH obviously increases after the introduction of SnO_2_, suggesting that the presence of SnO_2_ could accelerate the generation of ·OH. This is consistent with the results of the Fenton activity test. 

It has been reported that Fe^3+^ could act as an acceptor of photo-generated electrons from a semiconductor during the photocatalytic process to suppress electron–hole recombination [26,28,51,52]. The E_CB_ of SnO_2_ (0.4 eV) is more positive than the Fe^3+^/Fe^2+^ (0.77 eV) redox potential. Therefore, the photoelectrons from CB of SnO_2_ could transfer to the Fe^3+^ that are located at the abundant heterogeneous interfaces, and reduce Fe^3+^ to Fe^2+^. The Fe^2+^ could further react with H_2_O_2_ and produce more ·OH.

According to the above results, the catalytic mechanism was illustrated in Figure 8. Under visible light irradiation, the photoelectrons (e^–^) of the α-Fe_2_O_3_ are excited, and transition into the conduction band (CB), leaving the same amount of holes (h^+^) in the valence band (VB) [40]. The h^+^ and e^−^ can react with H_2_O_2_ to generate ·O_2_ and ·OH. The separation of h^+^ and e^−^ can be promoted owing to the presence of numerous H_2_O_2_ and Fe^3+^. Meanwhile, the generation of ·O_2_ and ·OH can also be accelerated [49,50]. Importantly, the CB of SnO_2_ could act as a sink for the generated electrons from the α-Fe_2_O_3_ and the excited MB molecules. Since the CB position of SnO_2_ is more negative than the Fe^3+^ redox potential, these electrons would be captured by the Fe^3+^ on the abundant interface of the SnO_2_/α-Fe_2_O_3_ heterogeneous catalyst, which can accelerate the cycle of Fe^3+^/Fe^2+^ and the photo-Fenton reaction for the generation of ·OH [51,52,53]. In this way, more ·OH and ·O_2_ radicals can be produced in SnO_2_/α-Fe_2_O_3_ compared with the α-Fe_2_O_3_, resulting in the significant enhancement of photocatalytic properties.

## 4. Conclusions

In summary, the x-SnO_2_/α-Fe_2_O_3_(x = 0.04, 0.07, and 0.1) heterogeneous catalysts were successfully prepared using a straightforward two-step hydrothermal strategy. The MB photodegradation investigation showed that the SnO_2_/α-Fe_2_O_3_ composites exhibited an excellent photodegradation ability, with the addition of H_2_O_2._ The rate constant of 0.07-SnO_2_/α-Fe_2_O_3_ composite (0.0537 min^−1^) is 2.8 times higher than that of pure α-Fe_2_O_3_ powder (0.0191 min^−1^). This remarkable enhancement is attributed to the effective transfer of photo-generated electrons for decomposing hydrogen peroxide into active radicals. The catalytic mechanism of the SnO_2_/α-Fe_2_O_3_ heterogeneous catalyst can provide a new insight into the catalytic mechanism of the photo-Fenton process.

## Data Availability

The data underlying this article will be shared on reasonable request from the corresponding author.

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
