# Peer review of "Enhanced Photo-Fenton Activity of SnO2/α-Fe2O3 Composites Prepared by a Two-Step Solvothermal Method"

_materials, 2022, doi:10.3390/ma15051743_

Round 1

Reviewer 1 Report

The article presented by the authors refers to a topic of general interest, composite materials made and tested can be a solution to replace conventional materials.

Experimental research is conducted based on a correct algorithm, the catalytic mechanism of photodegradation has been fully investigated, and the optimum for the rate constant is correct.

The figures presented correspond to the requirements of the journal in which the authors wish to publish this article, the large number of bibliographic references are considered and marked in the text.

I look forward to presenting your research on this topic.

Author Response

Thanks for your valuable suggestions and the recommendation to publish our manuscript.

Reviewer 2 Report

This work is based on the photo-Fenton activity of SnO2/α-Fe2O3 composites material for dye degradation is an already-reported material. However, the only advantage is decreased degradation time with the addition of hydrogen peroxide. This technique is also used by many research groups. Yet, it can be considered as a major revision after answering the following comments.

  1. The introduction must be elaborated mentioning other high-performance photocatalysts compared with that of SnO2/α-Fe2O3.
  2. FE-SEM image clarity should be enhanced. the EDAX pattern image is not clear. Authors should also provide a clearer TEM image with EDS mapping.
  3. XPS is not fitted well in Figure. 3b.
  4. The author should show the difference between dark absorption and sunlight irradiation in Figure 5a.
  5. Provide UV- visible absorption spectra of 07-SnO2 /a-Fe2O3 at dark, 0-60 min degradation of MB as it is an important criterion.
  6. Provide the cycle run tests to show the 0.07-SnO2/Fe2O3 catalyst stability. After the cycle run tests show the stability of the catalyst by SEM analysis.
  7. Calculate the bandgap for SnO2, α-Fe2O3, and SnO2/α-Fe2O3 from UV-DRS and explain the mechanism based on bandgap and energy difference. How the hole/electron is generated and how the free radical is formed in influencing the MB dye degradation.
  8. A similar type of work was also reported, comparing the performance of the prepared catalyst with the reports of Mater. Chem. Front., 2018, 2, 796-806, J. Phys. Chem. C 2011, 115, 7874-7879.

Author Response

The papers of Mater. Chem. Front., 2018, 2, 796-806, and J. Phys. Chem. C 2011, 115, 7874-7879 have been cited as Ref 41 and Ref 42 in the revision and the following descriptions are added:

For Ref 41 (Mater. Chem. Front., 2018, 2, 796-806) in the revision:The SnO2 and α-Fe2O3 were prepared by a sol–gel method and their composite (SnO2–α-Fe2O3) systems were synthesized by combining SnO2 with α-Fe2O3 in various weight percent ratios and the photocatalytic activity was investigated.

For Ref 42 (J. Phys. Chem. C 2011, 115, 16, 7874–7879) in the revision:A necklacelike SnO2/α-Fe2O3 hierarchical heterostructure was fabricated by chemical vapor deposition method using SnO2 nanowires with the preferential growth direction of [001] direction as template and photocatalytic property was studied. Please check lines 58-63 in the revision.

Our fabrication method differs to the above both literatures (sol–gel method for Ref 41 and chemical vapor deposition method for Ref 42). In particular, it should be noted both above literatures reported the photocatalytic property. In our study, it is focused on photo-Fenton activity. Both the physical mechanism and test method are different for photocatalytic property and photo-Fenton activity. So the analyzed techniques should be different.

Reviewer 3 Report

In this paper, the work done to prepare x-SnO2/α-Fe2O3 heterogeneous composites via two-step solvothermal method and to study its’ photo-Fenton activity. The steps taken for the preparation method were clearly defined. The results, analysis, and discussion were thoroughly explained, proving the worthiness of the composites for application in catalytic mechanism of photo-Fenton process. The cited references were mixed from old and recent works.

I have no major problem accepting the draft.   But I would like to enquire regarding line 127, is there any specific reason why the chosen wavelength peak for the absorbance was 664 nm. And if the author can include the explanation in the next manuscript version.

Author Response

Dear Referee 3,

We thank you very much for your valuable comments about our manuscript. We have revised the manuscript following your comments/suggestions carefully. All the revision/additions were marked with red color for your check. Please find our answer to your concerns point by point as follows:

General comment: In this paper, the work done to prepare x-SnO2/α-Fe2O3 heterogeneous composites via two-step solvothermal method and to study its’ photo-Fenton activity. The steps taken for the preparation method were clearly defined. The results, analysis, and discussion were thoroughly explained, proving the worthiness of the composites for application in catalytic mechanism of photo-Fenton process. The cited references were mixed from old and recent works.

Response: Thanks for your valuable suggestions and the recommendation to publish our manuscript.

Comment: I have no major problem accepting the draft. But I would like to enquire regarding line 127, is there any specific reason why the chosen wavelength peak for the absorbance was 664 nm. And if the author can include the explanation in the next manuscript version.

Response: Thanks for your suggestions. For a clearer description, “where the solution has the maximum absorption.” was added in line 136 in the revision, please check.

We sincerely appreciate your time and effort to check our manuscript again.

Best regards

Instead of all co-authors for this submission by:

Xianmin Zhang, Ph.D. 

Reviewer 4 Report

The presented manuscript seems to be interesting for readers of the Materials journal, it is written in a good manner and suits the requirements of the journal. It can be accepted for publication after minor corrections listed below.

- Section 2.1. Materials and chemicals Lines 76 to 78 must be given a complete sentence.

- The purity of the materials used must be stated.

- More details of the process should be given in Scheme 1.Doing this, review and citing the following ref could be helpful:  Materials Research Express 7, no. 4 (2020): 045008.

- The reason for choosing 0.04, 0.07 and 0.1 for SnO2 should be given with more evidence

-It is suggested to bring the Results and discussion section with an independent number, i.e.” 3. Results and discussion”.

- Regarding the synthesized particle size, it should be calculated and discussed between the XRD and FESEM results. For this purpose, it is recommended to study and refer to the appropriate source to calculate the crystallite size and particle size.

Author Response

Dear Referee 4,

We thank you very much for your valuable comments about our manuscript. We have revised the manuscript following your comments/suggestions carefully. All the revision/additions were marked with red color for your check.

General comment: The presented manuscript seems to be interesting for readers of the Materials journal, it is written in a good manner and suits the requirements of the journal. It can be accepted for publication after minor corrections listed below.

Response: Thanks for your valuable suggestions and the recommendation to publish our manuscript.

Comment 1: Section 2.1. Materials and chemicals Lines 76 to 78 must be given a complete sentence.

Response 1: Thanks for your suggestions. The description has been changed to “Ferric chloride hexahydrate (FeCl3·6H2O), tin chloride pentahydrate (SnCl4·5H2O), urea (CO(NH2)2), polyethylene glycol (PEG), sodium hydroxide (NaOH), ethanol (C2H5OH) and methylene blue (MB) were used in the experiment.” in the revision. Please check lines 82-84.

Comment 2: The purity of the materials used must be stated.

Response 2: Thanks for your suggestions. “All the reagents were analytical grade and used as received without further purification.” has been stated in our first manuscript. Please check lines 84 and 85.

Comment 3: More details of the process should be given in Scheme 1. Doing this, review and citing the following ref could be helpful:  Materials Research Express 7, no. 4 (2020): 045008.

Response 3: Thanks for your suggestions. The paper of [Mater. Res. Express. 7 (2020) 045008] was cited as Ref 43 and the description “The precise control of reaction temperatures is very important to obtain the target samples with good crystal structures and homogeneous particles size [43]”. Please check lines 109-111.

Comment 4: The reason for choosing 0.04, 0.07 and 0.1 for SnO2 should be given with more evidence.

Response 4: Thanks for your suggestions. By quantitative calculation, we chose x= 0.04, 0.7 and 0.1 to make the mass contents of Fe2O3 in x-SnO2/α-Fe2O3 between 70% and 90%.

Comment 5: It is suggested to bring the Results and discussion section with an independent number, i.e.” 3. Results and discussion”.

Response 5: Thanks for your suggestions. The “2.5 Results and discussion” part in the first manuscript has been changed to “23. Results and discussion” part in the revision. Please check in line 137.The “3. Conclusion” part has been changed to “4. Conclusion” Part in line 279.

Comment 6: Regarding the synthesized particle size, it should be calculated and discussed between the XRD and FESEM results. For this purpose, it is recommended to study and refer to the appropriate source to calculate the crystallite size and particle size.

Response 6: Thanks for your suggestions. In the revision, the particle size was estimated using the Scherrer’s equation [X. Ai et al. Applied Surface Science 428 (2018) 788–792 as Ref 44]

The particle sizes basically agree with the results from the SEM as shown in Figure 2a. Please check lines 147-150.

We sincerely appreciate your time and effort to check our manuscript again.

Best regards

Instead of all co-authors for this submission by:

Xianmin Zhang, Ph.D.

Reviewer 5 Report

The manuscript was aimed at Enhancing photo-Fenton activity of SnO2/α-Fe2O3 composites, highlighting interesting results. However, minor comments listed below should be addressed:

  1. Abstract: the aim of this study should be stated.
  2. The novelty of this work should be emphasized at the end of the introduction to highlight the gap between the current study and previous studies.
  3. Have the authors performed statistical analyses of the obtained results? Please add a section for statistical analyses, demonstrating the employed software and adopted tests at the end of the methodology. Additionally, how many replicates have been done over the entire experiment? Please follow this article for this purpose and you could cite it (https://doi.org/10.3390/ijms222313050).
  4. Fig. 2: I could not view the results of the EDX chart, please improve it or add the results in a table.
  5. Fig. 5 and 6: please redraw these figures using means of at least three replicates with SD or SEM.

Author Response

In the revision, the result of the EDX chart was individually shown in Figure 2c to make it more clear.

The paper of [titled: Development of Polyvinyl Alcohol/Kaolin Sponges Stimulated by Marjoram as Hemostatic, Antibacterial, and Antioxidant Dressings for Wound Healing Promotion, published in [Int. J. Mol. Sci. 2021, 22(23), 13050] (https://doi.org/10.3390/ijms222313050)] has no relation with our present study. So we can not cite it and fell other suggestions are not suitable for the present study.

Round 2

Reviewer 2 Report

A proper response letter should be provided. A prompt reply is needed for all the questions.

Authors should check again the English (minor spell check required). Example-Line 170-EDS spectra or spectrum.

Reviewer 4 Report

As authors have performed an adequate revise, the manuscript might be accepted for publication in the journal of Materials.